# Screening and Identification of Key Biomarkers in Metastatic Uveal Melanoma: Evidence from a Bioinformatic Analysis

**DOI:** 10.3390/jcm11237224

**Published:** 2022-12-05

**Authors:** Tan Wang, Zixing Wang, Jingyuan Yang, Youxin Chen, Hanyi Min

**Affiliations:** 1Department of Ophthalmology, Peking Union Medical College Hospital, Chinese Academy of Medical Sciences and Peking Union Medical College, Beijing 100730, China; 2Key Laboratory of Ocular Fundus Diseases, Chinese Academy of Medical Sciences and Peking Union Medical College, Beijing 100730, China; 3Institute of Basic Medical Sciences, Chinese Academy of Medical Sciences/School of Basic Medicine, Peking Union Medical College, Beijing 100730, China

**Keywords:** uveal melanoma, biomarkers, microarray, bioinformatics, ROC curve, survival analysis

## Abstract

Purpose: To identify key biomarkers in the metastasis of uveal melanoma (UM). Methods: The microarray datasets GSE27831 and GSE22138 were downloaded from the Gene Expression Omnibus database. Differentially expressed genes (DEGs) were identified, and functional enrichment analyses were performed. A protein–protein interaction network was constructed, and four algorithms were performed to increase the reliability of hub genes. Biomarker analysis and metastasis-free survival analysis were performed to screen and verify prognostic hub genes. Results: A total of 138 DEGs were identified, consisting of 71 downregulated genes and 67 upregulated genes. Four genes (ROBO1, FMN1, FYN and FXR1) were selected as hub genes. Biomarker analysis and metastasis-free survival analysis showed that ROBO1, FMN1, FYN and FXR1 were factors affecting the metastasis and metastasis-free survival of UM (all *p* < 0.05). High expression of ROBO1 and low expression of FMN1 were associated with longer metastasis-free survival. Multivariable logistic regression and Cox analyses in GSE 27831 indicated that ROBO1 was an independent factor affecting metastasis and metastasis-free survival of UM (*p* = 0.010 and *p* = 0.009), while ROBO1 and FMN1 were independent factors affecting metastasis and metastasis-free survival of UM in GSE22138 (all *p* < 0.05). Conclusions: ROBO1, FMN1, FYN and FXR1 should be regarded as diagnostic biomarkers for the metastasis of UM, especially ROBO1 and FMN1. High expression of ROBO1 and low expression of FMN1 were associated with longer metastasis-free survival. This study may facilitate the understanding of the molecular mechanisms underlying the metastasis of UM.

## 1. Introduction

The most common type of intraocular cancer among adults is uveal melanoma (UM). Metastases develop in up to 50% of patients within 36 months [1,2]. Once metastasis is identified, the patient survival rate rapidly drops to approximately 15% after one year, with a median survival time estimated to be between 4 and 15 months [3]. Moreover, there is currently no efficient treatment for patients with metastasis. Therefore, early screening of patients at high risk and early diagnosis of metastases are crucial. Creating risk stratification and developing individualized follow-up plans can save a substantial amount of medical resources, minimize the cost of unnecessary invasive examinations and testing and result in a reasonable allocation of medical resources.

Over the past few decades, microarray technology and bioinformatic analyses have been widely used to screen genetic alterations at the genome level, enabling the identification of differentially expressed genes (DEGs) and functional pathways involved in the carcinogenesis and metastasis of UM. A number of studies have identified genes and pathways that have contributed to advancements in the diagnosis and treatment of UM [4,5]. Nonetheless, the prevalence of false-positive outcomes in independent microarray analysis makes it challenging to acquire trustworthy data, and few effective biomarkers have been identified for the assessment of carcinogenesis and metastasis in UM.

Thus, in the present study, two mRNA microarray datasets were acquired from Gene Expression Omnibus (GEO) and analyzed in order to identify DEGs between metastatic and nonmetastatic samples. To understand the molecular mechanisms driving carcinogenesis and metastasis, pathway and process enrichment analysis and protein–protein interaction (PPI) network analyses were then performed. A total of 138 DEGs and 4 hub genes were identified, which may be candidate biomarkers for metastasis in UM. Additionally, biomarker analysis and metastasis-free survival analysis of the hub genes were performed in order to validate the role of hub genes in predicting metastasis.

## 2. Methods

### 2.1. Microarray Data

GEO (http://www.ncbi.nlm.nih.gov/geo (accessed on 2 May 2022)) [6] is a public functional genomics data repository of high-throughput gene expression data, chips and microarrays.

The search strategy (‘uveal melanoma’ [MeSH Terms] AND (‘Homo sapiens’ [Organism] AND ‘Expression profiling by array’ [Filter]) was adopted.

Inclusion criteria were as follows: [1] UM with metastasis as test samples and [2] UM displaying no metastasis as controls. We extracted the gene expression datasets GSE27831 [7] and GSE22138 [8] from the GEO database. The platform for GSE27831 was GPL570 [HG-133_Plus_2] Affymetrix Human Genome U133 Plus 2.0 Array, which included 11 metastatic samples and 18 nonmetastatic samples. The platform for GSE22138 was GPL570, [HG-U133_Plus_2] Affymetrix Human Genome U133 Plus 2.0 Array, which contained 35 metastatic samples and 28 nonmetastatic samples. Data table header descriptions and series matrix files of GSE27831 and GSE22138 were downloaded. The probes were converted into the corresponding gene symbol according to the annotation information in the platform.

### 2.2. Identification of DEGs

The DEGs between metastatic samples and nonmetastatic samples were screened using GEO2R (http://www.ncbi.nlm.nih.gov/geo/geo2r (accessed on 2 May 2022)). GEO2R is an interactive web tool that allows users to compare two or more datasets in a GEO series to identify DEGs across experimental conditions. The adjusted *p* values (adj. *p*) and Benjamini and Hochberg false discovery rates were used to apply adjustment to the *p* values. Log transformation and force normalization were performed. Probe sets without corresponding gene symbols or genes with more than one probe set were removed. |LogFC| > 0.58 and *p* < 0.01 were considered to be statistically significant. The online tool jvenn (http://jvenn.toulouse.inra.fr/app/index.html (accessed on 2 May 2022)) was used to detect common DEGs among the two datasets.

### 2.3. Pathway and Process Enrichment Analysis of DEGs

GO analysis has been used extensively to identify the characteristic biological attributes of genes, gene products and sequences, including the biological process (BP), cell components (CC) and molecular function (MF) [9]. KEGG analysis provides a comprehensive biointerpretation of genomic sequences and protein interaction network information [10].

In this study, GO terms and KEGG pathway enrichment analysis of DEGs were automatically completed and visualized using clusterProfiler V3.14.0 [11], pathview V1.36.0 [12] and the Goplot V1.0.2 package [13] in the R software statistical analysis platform (significance was *p* < 0.05 and a *q*-value < 0.05).

### 2.4. PPI Network Construction and Hub Gene Identification 

The PPI network of the differentially coexpressed genes was established using the Search Tool for the Retrieval of Interacting Genes (STRING) [14]. Cytoscape was used to build the visual network of molecular interactions with a combined score > 0.15 [15]. The degree, edge percolated component (EPC), maximal clique centrality algorithm (MCC) and maximum neighborhood component (MNC) algorithms were used to select hub genes from the PPI networks [16]. We calculated the degree, EPC, MCC and MNC scores of all nodes of the PPI network via the CytoHubba plugin. The top 10 nodes with the highest degree, EPC, MCC and MNC scores were selected, and we took the intersection of the outcomes of the four algorithms. To increase the reliability of hub genes, their overlapping genes were established as hub genes.

### 2.5. Statistical Analysis

Regarding baseline clinical characteristics, the normal distribution of all variables was verified by the Kolmogorov–Smirnov method. The independent Student’s *t* test or Mann–Whitney U test was used to compare continuous variables, whereas the chi-square test or Fisher’s exact test was applied to compare categorical data.

Univariable and multivariable logistic regression analyses were conducted for the hub genes in GSE27831 and GSE22138. Then, the receiver operating characteristic (ROC) curves of the hub genes were examined for metastasis, and the area under the curve (AUC) of each ROC curve was calculated.

Each dataset was divided into two groups using the median of the hub gene expression levels as the boundary (the high-expression group and the low-expression group). Metastasis-free survival analyses of the hub genes were performed using Kaplan–Meier curves and the log-rank test to explore how these genes affect metastasis-free survival. Univariate Cox regression analysis was used to screen prognostic hub genes, and a multivariate Cox proportional hazards regression model was used to screen for independent prognostic hub genes.

A *p* value of less than 0.05 was considered to be statistically significant. All statistical analyses were performed using SPSS software (version 22.0; SPSS Inc., Chicago, IL, USA).

## 3. Results

### 3.1. Clinical Characteristics

Patient characteristics in the GSE27831 and GSE22138 cohorts are given in Table 1. There were no significant differences between the two cohorts in metastasis prevalence (*p*= 0.116). Metastasis positivity was 37.9% and 55.6% in the GSE27831 and GSE22138 cohorts, respectively. There were significant differences in the proportion monosomy of chromosome 3 in the GSE27831 and GSE22138 cohorts (*p* = 0.019 and *p* = 0.002, respectively).

### 3.2. Identification of DEGs

After standardization of the datasets (Appendix A), 846 and 666 DEGs were extracted from GSE27831 and GSE22138, respectively, based on the defined criteria. The DEGs are shown in the volcano plots and the heatmaps (Figure 1), of which GSE27831 included 400 upregulated genes and 446 downregulated genes, and GSE22138 included 370 upregulated genes and 296 downregulated genes. The coexpressed DEGs were integrated using the Venn Diagram online tool and included 67 upregulated and 71 downregulated genes (Appendix A).

### 3.3. Pathway and Process Enrichment Analysis of DEGs

The enriched GO and KEGG pathways of 138 coexpressed DEG genes were analyzed and visualized (Figure 2 and Figure 3). The GO enrichment results indicated that for BP, DEGs were significantly enriched in response to acid chemical, water-soluble vitamin metabolic process, folic acid metabolic process, glycoprotein biosynthetic process, modified amino acid transport, folic-acid-containing compound metabolic process, response to amino acid, IRE1-mediated unfolded protein response, cellular response to amino acid stimulus and endoplasmic reticulum unfolded protein response. In terms of CC, DEGs were significantly enriched in the microtubule, costamere, Golgi apparatus subcompartment, organelle subcompartment, phagocytic cup, cytoplasmic microtubule, podosome, trans-Golgi network, dendritic spine and neuron spine. For MF, DEGs were significantly enriched in modified amino acid transmembrane transporter activity, glucuronosyltransferase activity, protease binding, dicarboxylic acid transmembrane transporter activity, acetylgalactosaminyltransferase activity, SH3 domain binding, actin monomer binding, UDP-glycosyltransferase activity, vitamin transmembrane transporter activity and neutral amino acid transmembrane transporter activity. The KEGG pathway enrichment results indicated that DEGs were mainly enriched in glycosaminoglycan biosynthesis—chondroitin sulfate/dermatan sulfate, focal adhesion, cysteine and methionine metabolism and glutathione metabolism.

### 3.4. PPI Network Construction and Hub Gene Identification

The PPI network of the 138 DEGs with 75 nodes and 416 edges is depicted in Figure 4. The degree, EPC, MCC and MNC scores of DEGs were calculated through the CytoHubba plugin, and we selected the ten genes with the highest scores in each algorithm and then took the intersection of the four groups to improve the reliability of hub genes. Finally, a total of four genes (ROBO1, FMN1, FYN and FXR1) were considered to be hub genes (Figure 4 and Appendix A).

### 3.5. Biomarker Analysis of the Hub Genes

In the two cohorts, univariable logistic regression analysis showed that ROBO1, FMN1, FYN and FXR1 were prognostic factors for metastasis (all *p* < 0.05). Then, multivariable logistic regression analysis was performed using these four genes (ROBO1, FMN1, FYN and FXR1). Multivariable logistic regression analysis in GSE 27831 indicated that ROBO1 was an independent factor affecting metastasis (*p* = 0.010), while ROBO1 and FMN1 were independent factors affecting metastasis in GSE22138 (*p* = 0.023 and *p* = 0.047, respectively; Table 2).

As shown in Table 3 and Figure 5, ROC curve analyses were performed in order to examine risk factors for metastasis. ROBO1, FMN1, FYN and FXR1 all achieved statistical significance for classifying metastasis status in GSE 27831 and GSE22138 (all *p* < 0.05). ROBO1 showed the largest AUC in GSE27831 (0.904). FMN1 showed the largest AUC in GSE22138 (0.780). Combining ROBO1 and FMN1 in GSE22138 showed that the AUC was 0.821 (*p* < 0.001).

### 3.6. Metastasis-Free Survival Analysis of the Hub Genes

Each dataset was divided into two groups using the median of the hub gene expression levels as the boundary (the high-expression group and the low-expression group). As shown in Figure 6, the Kaplan–Meier curves of GSE27831 and GSE22138 showed that the metastasis-free survival of the ROBO1 high-expression group was significantly higher than that of the ROBO1 low-expression group (*p* = 0.003 and *p* < 0.001 for GSE27831 and GSE22138, respectively); the metastasis-free survival of the FYN high-expression group was significantly higher than that of the FYN low-expression group (*p* = 0.006 and *p* = 0.002 for GSE27831 and GSE22138, respectively); the metastasis-free survival of the FXR1 high-expression group was significantly higher than that of the FXR1 low-expression group (*p* = 0.018 and *p* = 0.016 for GSE27831 and GSE22138, respectively); and the metastasis-free survival of the FMN1 low-expression group was significantly higher than that of the FMN1 high-expression group (all *p* < 0.001). The median metastasis-free survival times of each hub gene in the high-expression group and the low-expression group are shown in Table 4.

In the two cohorts, univariable Cox regression analysis showed that ROBO1, FMN1, FYN and FXR1 were all prognostic factors for metastasis-free survival (all *p* < 0.05). Then, multivariable Cox regression analysis was performed using those four genes. Multivariable Cox regression analysis in GSE27831 indicated that ROBO1 was an independent factor affecting metastasis-free survival (*p* = 0.009), while ROBO1 and FMN1 were independent factors affecting metastasis-free survival in GSE22138 (*p* = 0.007 and *p* < 0.001, respectively; Table 5).

## 4. Discussion

UM is the most prevalent intraocular primary tumor in adults, affecting the choroid, iris and ciliary body and has a strong propensity to metastasize, resulting in a high mortality rate [17,18,19]. The pathogenesis of melanoma is known to be highly complex and diverse [20]. Exposure to ultraviolet light has been demonstrated to be a leading cause of melanoma. Ultraviolet radiation induces a range of gene alterations, including those in BRAF, RAS, C-Kit and NF1, and promotes the activation of inflammation in melanoma, according to accumulating evidence [3]. However, the exact mechanism of the development and metastasis of UM remains unclear, and effective biomarkers to assess carcinogenesis and metastasis are lacking. In recent years, microarray technology has enabled the investigation of genetic mutations and has proven to be an effective method for identifying new biomarkers in different diseases.

In this study, two mRNA microarray datasets from GEO were analyzed to obtain DEGs between metastatic samples and nonmetastatic samples. A total of 138 DEGs were identified among the two datasets, including 71 downregulated genes and 67 upregulated genes. Pathway and process enrichment analyses were carried out among the DEGs. Many of the functions and pathways in the results of this study have been previously reported to be associated with cancer metastasis or development, including the folic acid metabolic process [21], glycoprotein biosynthetic process [22], glycosaminoglycan biosynthesis—chondroitin sulfate/dermatan sulfate [23], glucuronosyltransferase activity [24], acetylgalactosaminyltransferase activity [25], focal adhesion, cysteine and methionine metabolism [26] and glutathione metabolism [27]. 

The ten genes among the DEGs with the highest scores in each algorithm were selected, including the degree, EPC, MCC and MNC algorithms, and the intersection of the results of the four groups was analyzed in order to determine four genes (ROBO1, FMN1, FYN and FXR1) as hub genes.

Roundabout (ROBO) axon guidance molecules perform crucial functions in the development of numerous organs and tissues, such as the brain, lung and breast, and govern cell migration and death [28,29,30,31]. ROBO1 is known to be expressed in fetal tissues, particularly the nervous system, and was first identified as a tumor-specific antigen in liver cancer in 2006 [32]. Several studies have suggested that, in most cancers, ROBO proteins are downregulated, such as in lung cancer [33,34,35], brain cancer [33,36], cervical cancer [37], liver cancer [34,36] and colon cancer [35], among others. It has been reported that ROBO1 plays an important role in cancer invasion, migration, epithelial–mesenchymal transition and tumor-induced angiogenesis through SLIT2/ROBO1 signaling [38,39,40]. The SLIT2/ROBO1 pathway inhibits cell invasion by interacting with E-cadherin and β-catenin in breast cancer and colorectal cancer [41,42,43], whereas in liver cancer, SLIT2/ROBO1 specifically inhibits hepatocyte growth factor (HGF)–mediated cell migration [42]. These findings indicate that ROBO1 mainly inhibits tumor progression, invasion, migration and apoptosis through the SLIT2/ROBO1 pathway.

In the current study, the ROBO1 gene was downregulated in metastatic samples and was suggested to be a hub gene in all DEGs between metastatic samples and nonmetastatic samples. Multivariable logistic regression and multivariate Cox proportional hazards regression analysis indicated that ROBO1 was an independent factor affecting metastasis and metastasis-free survival. The ROC curve showed that ROBO1 was an eligible biomarker to predict metastasis. Moreover, the Kaplan–Meier curves showed that high expression of ROBO1 was associated with longer metastasis-free survival. These data, combined with the results of previous studies, indicate that ROBO1 may play a crucial role in the metastasis of UM.

Formin-1 (FMN1) is the founding formin family member [44,45]. It is an authentic formin capable of nucleating actin filaments in vitro [46]. FMN1 is involved in cell–cell adhesion [46], focal adhesion formation in primary epithelial cells [47], dendritogenesis and synaptogenesis in hippocampal cultures [47,48]. It has been reported that FMN1 expression is associated with motility in different cancer cell lines, and its ectopic expression has been shown to boost fitness indices. Specifically, FMN1 is able to generate actin filaments from the cytosol and microtubule lattice. This could improve the density of the cytoskeleton gel and the mechanical cohesiveness that facilitates cell migration and direction shift. FMN1 generates robust mechanical cohesion by acting on the microtubule lattice, resulting in highly invasive motility.

We also found that FMN1 was related to metastasis in UM. Multivariable logistic regression and Cox regression analysis indicated that FMN1 was an independent factor affecting metastasis and metastasis-free survival in GSE22138. The ROC curve showed that FMN1 was also an eligible biomarker to predict metastasis. Moreover, the Kaplan–Meier curves showed that low expression of FMN1 was associated with longer metastasis-free survival. Thus, FMN1 might act as an important gene to facilitate the migration of UM.

FYN proto-oncogene, Src family tyrosine kinase (FYN) is a 59-kDa protein-encoding gene that is implicated in cell growth, survival, cell motility and adhesion [49,50]. Several studies have revealed that FYN plays crucial functions in a variety of malignancies. Lee et al. found that FYN establishes a positive feedback loop with STAT5 to promote breast cancer cell metastasis through NOTCH2 activation [51]. Another group demonstrated that tumorigenesis induced by the depletion of PTPN23 can be reversed by the suppression of FYN or through the Src inhibitor AZD0530 [52]. A study of pancreatic cancer also demonstrated that FYN inhibition promotes the phosphorylation and nuclear localization of hnRNP E1, which ultimately suppresses pancreatic cancer cell metastasis and invasion [53]. Glioblastoma research has revealed that PIKE-A impairs the tumor suppressive actions of AMPK, which are mediated by FYN [54]. In hepatocellular carcinoma, FYN-mediated activation of the STAT3 pathway plays an important role in Fzd2-driven EMT and the migration of liver cancer cells [55]. In gastric cancer, FYN was overexpressed and positively correlated with metastasis. FYN knockdown significantly decreased cancer cell migration and invasion, whereas FYN overexpression increased cancer migration and invasion. Genetic inhibition of FYN decreased the number of metastatic lung nodules in vivo.

In the present study, the FYN gene was upregulated in metastatic samples, and the degree, EPC, MCC and MNC scores of the CytoHubba plugin suggested it as a hub gene in all DEGs. Univariable logistic regression and Cox regression analysis indicated that FYN was a prognostic factor affecting metastasis and metastasis-free survival. The ROC curve showed that FYN was also an eligible biomarker to predict metastasis. Taken together, the data suggest that FYN may act as a crucial oncogene that could promote the migration and invasion of UM.

Fragile X mental retardation-associated protein 1 (FXR1) is a highly conserved cytoplasmic RNA-binding protein among vertebrates. It has been studied for its role in muscle development, inflammation and tumorigenesis. FXR1 has been investigated in oncology for its potential role as a key regulator of tumor progression. It has been observed to be overexpressed at the genetic level, for example, in human and canine uveal malignant melanoma [56,57], human lung squamous cell carcinoma, non-small-cell lung cancer [58,59], rhabdomyosarcoma [60], head and neck squamous cell carcinomas [60], glioma [61], prostate cancer and colorectal cancer [62]. Downregulation of FXR1 can result in the inhibition of glioma cell progression [63]. In prostate cancer cells, FXR1 negatively regulated FBXO4 transcripts via direct association with its 3’UTR and promoted mRNA degradation. FBXO4 knockdown predominantly rescued the tumor-suppressive phenotype in FXR1-deficient cells [62]. 

The results of the present study showed overexpression of FXR1 at the genetic level in metastatic samples. The degree, EPC, MCC and MNC scores of the CytoHubba plugin suggested it as a hub gene in all DEGs. Univariable logistic regression and Cox regression analysis indicated that FYN was a prognostic factor affecting metastasis and metastasis-free survival. The ROC curve showed that FYN was also an eligible biomarker to predict metastasis. All these data indicate that FXR1 may act as a promoter of tumor progression and metastasis, including in UM.

There are some limitations of this study. First, although we used two external databases, these were all single-omics data of gene expression. Compared with multiomics studies, the exploration of the mechanism was not comprehensive enough. Second, we did not perform functional experiments on hub genes in vivo or in vitro. Therefore, the specific mechanism of hub genes in UM progression and metastasis has not been demonstrated. In the future, combining these biomarkers with clinical indicators and radiomics to establish an ideal prediction model will be conducive to the early prediction of metastasis and metastasis-free survival. In addition, we will also use these four hub genes in our future research to investigate whether they can be used as therapeutic targets. 

In conclusion, a total of 138 DEGs and 4 hub genes were identified and may be regarded as diagnostic biomarkers for the metastasis of UM, especially ROBO1 and FMN1. High expression of ROBO1 and low expression of FMN1 were associated with longer metastasis-free survival.

## Figures and Tables

**Figure 1 jcm-11-07224-f001:**
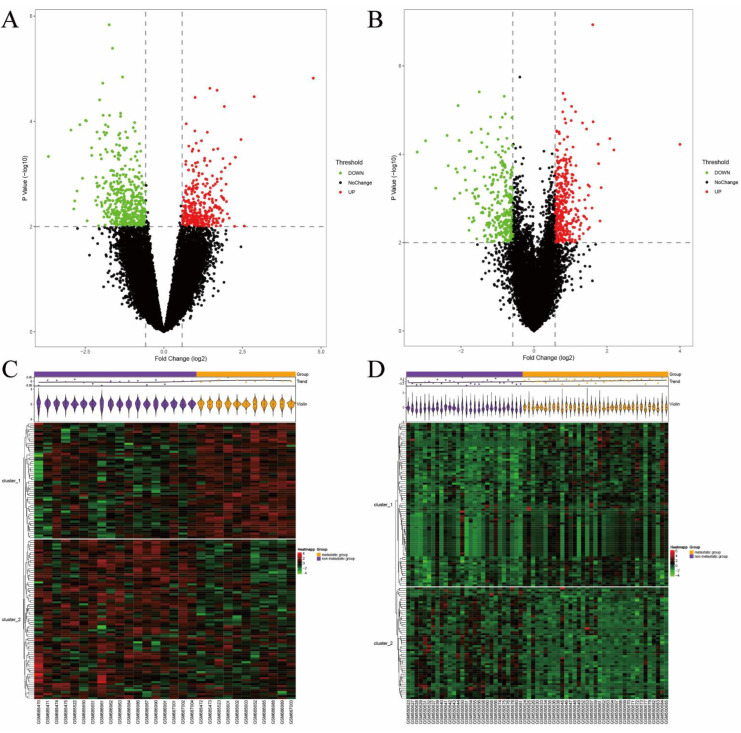
Volcano plots of differentially expressed genes in (**A**) GSE27831 and (**B**) GSE22138. Data points in red represent upregulated genes, and green represents downregulated genes. The differences were set as *p* < 0.01 and |log FC| > 0.58. Heatmap of differentially expressed genes identified in (**C**) GSE27831 and (**D**) GSE22138. The legend on the top right indicates the log fold change of the genes.

**Figure 2 jcm-11-07224-f002:**
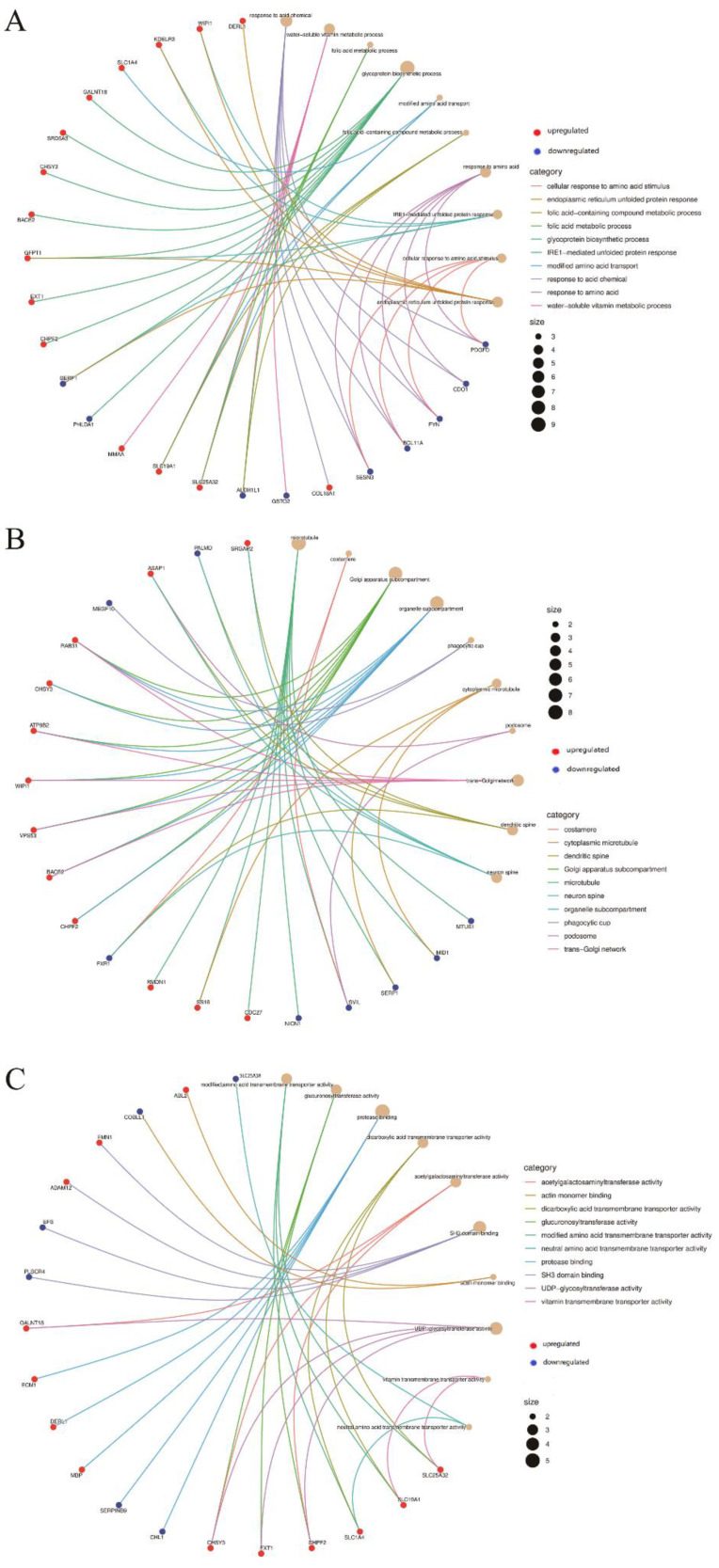
Gene Ontology (GO) enrichment analysis of differentially expressed genes (DEGs). Cnetplot shows GO enrichment significance items of DEGs in three functional groups: (**A**) molecular function (MF), (**B**) biological processes (BP) and (**C**) cell composition (CC). Symbols of DEGs are presented on the left side of the graph. Symbols in red represent upregulated genes, and blue represents downregulated genes. Gene involvement in the GO terms is indicated by colored connecting lines.

**Figure 3 jcm-11-07224-f003:**
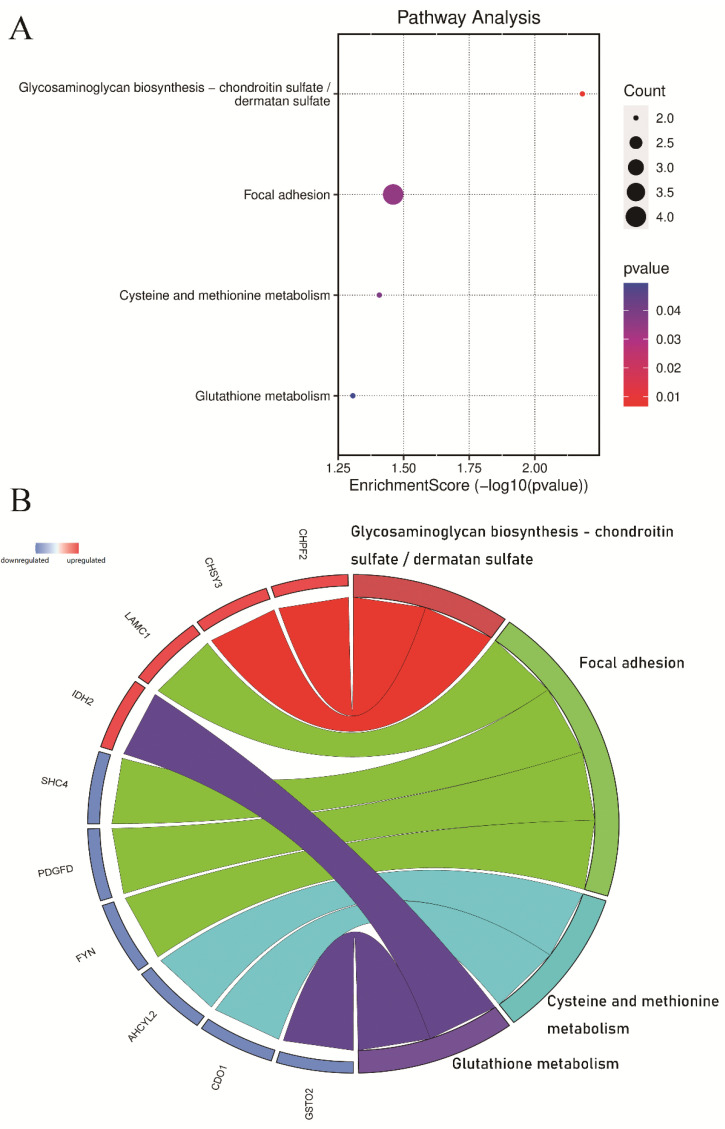
Kyoto Encyclopedia of Genes and Genomes (KEGG) pathway analysis of DEGs. (**A**) Advanced bubble chart shows enrichment of DEGs in signaling pathways. The *x*-axis label represents the gene ratio, and the *y*-axis label represents the pathway. (**B**) Chord plot shows the distribution of DEGs in different KEGG pathways. Symbols of DEGs are presented on the left side of the graph. Symbols in red represent upregulated genes, and blue represents downregulated genes. Gene involvement in the KEGG pathways is indicated by colored connecting lines.

**Figure 4 jcm-11-07224-f004:**
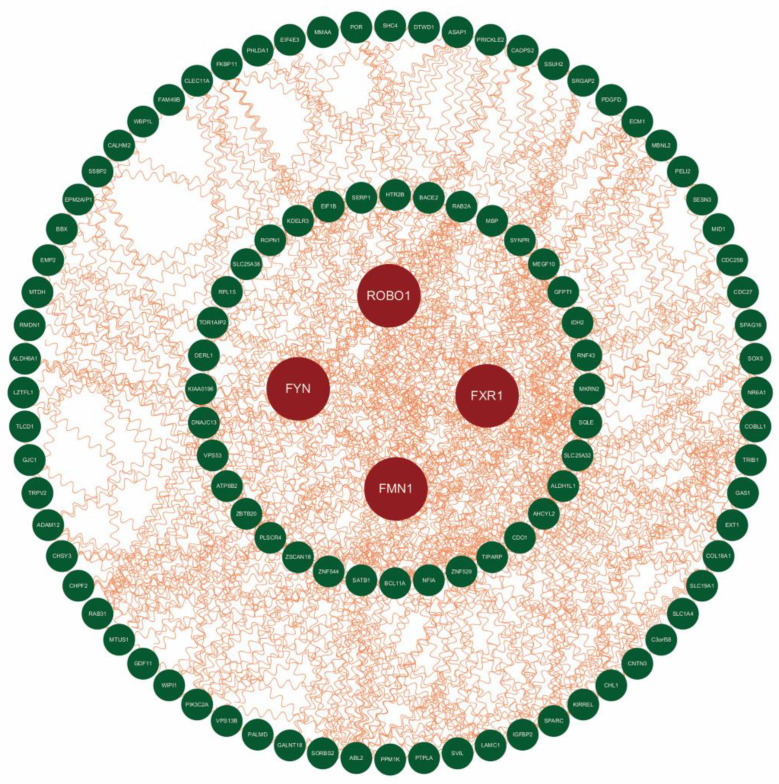
Protein–protein interaction (PPI) network of differentially expressed genes. Red represents the hub genes we selected.

**Figure 5 jcm-11-07224-f005:**
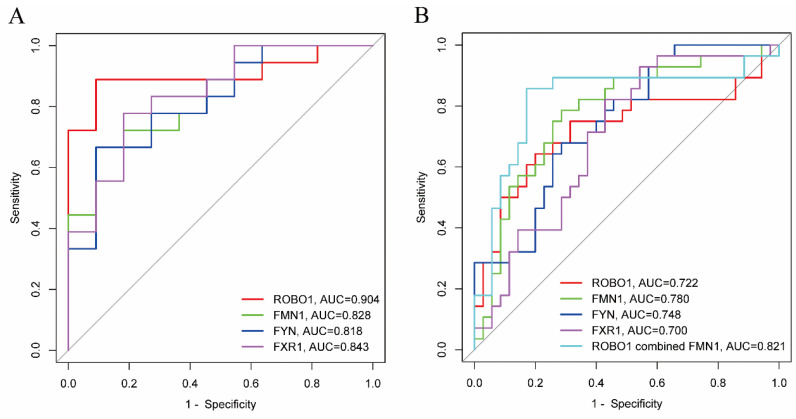
ROC curves of hub genes predicting metastasis. (**A**) ROC curves of ROBO1, FMN1, FYN and FXR1 in GSE 27831. (**B**) ROC curves of ROBO1, FMN1, FYN, FXR1 and ROBO1 combined with FMN1 in GSE22138.

**Figure 6 jcm-11-07224-f006:**
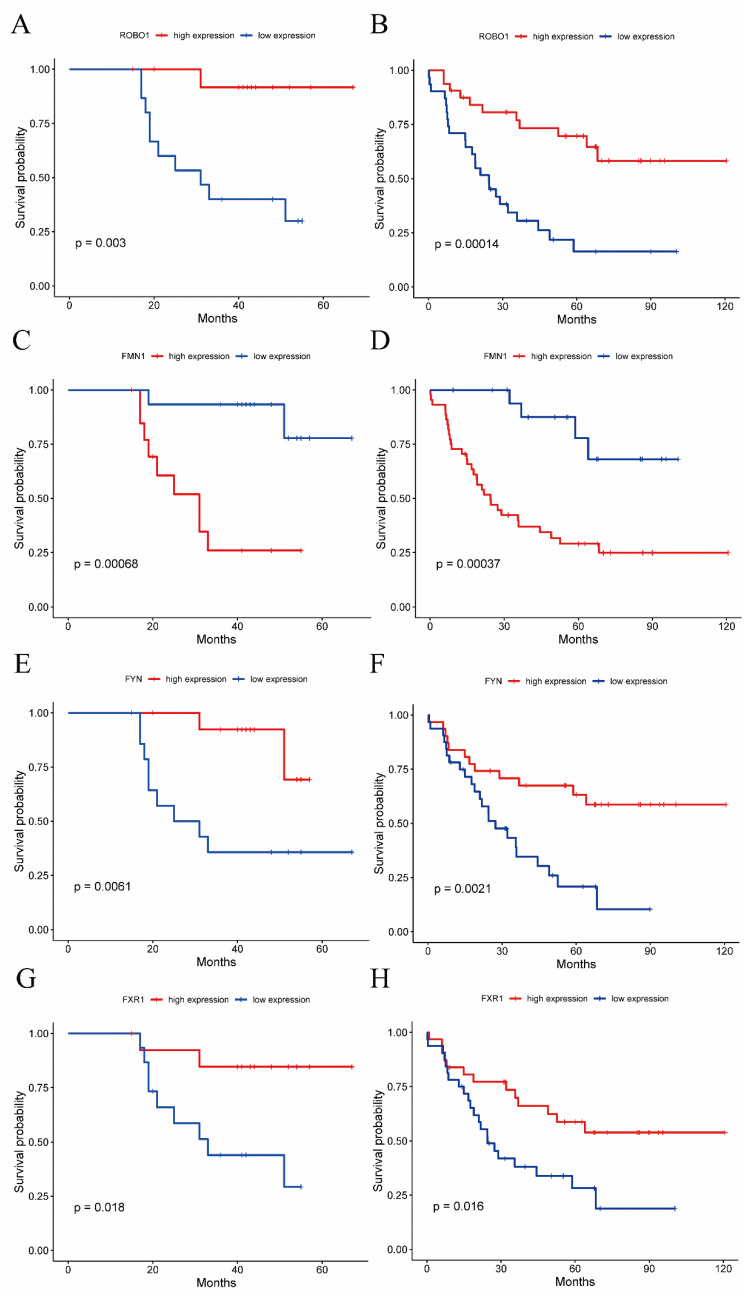
Kaplan–Meier curves of GSE27831 and GSE22138. (**A**) Kaplan–Meier curves of GSE27831, divided into two groups using the median ROBO1 expression levels as the boundary. (**B**) Kaplan–Meier curves of GSE22138, divided into two groups using the median ROBO1 expression levels as the boundary. (**C**) Kaplan–Meier curves of GSE27831, divided into two groups using the median FMN1 expression levels as the boundary. (**D**) Kaplan–Meier curves of GSE22138, divided into two groups using the median FMN1 expression levels as the boundary. (**E**) Kaplan–Meier curves of GSE27831, divided into two groups using the median FYN expression levels as the boundary. (**F**) Kaplan–Meier curves of GSE22138, divided into two groups using the median FYN expression levels as the boundary. (**G**) Kaplan–Meier curves of GSE27831, divided into two groups using the median FXR1 expression levels as the boundary. (**H**) Kaplan–Meier curves of GSE22138, divided into two groups using the median FXR1 expression levels as the boundary.

**Table 1 jcm-11-07224-t001:** Baseline characteristics of subjects included in the analysis *.

		GSE27831			GSE22138		
Variables		Metastatic Group (n = 11)	Nonmetastatic Group (n = 18)	*p* ^†^	Metastatic Group (n = 35)	Nonmetastatic Group (n = 28)	*p* ^†^
Age, y		66.7 ± 12.3	65.6 ± 13.4	0.823 ^t^	62.5 ± 9.6	59.1 ± 15.0	0.306 ^t^
Male gender (%)		7 (63.6)	10 (55.6)	0.717 ^F^	22 (62.9)	17 (60.7)	0.862 ^P^
Tumor location	anterior	2 (18.2)	2 (12.5)	0.653 ^F^	2 (5.9)	1 (4.2)	0.305 ^F^
	middle	5 (45.5)	10 (62.5)		22 (64.7)	20 (83.3)	
	posterior	4 (36.4)	4 (25.0)		6 (17.6)	3 (12.5)	
	2 or 3 locations				4 (11.8)	0 (0)	
Tumor diameter (mm)		13.0 ± 4.0	13.0 ± 6.0	0. 912 ^U^	15.2 ± 3.7	15.6 ± 3.9	0.921 ^t^
Tumor thickness (mm)		9.9 ± 4.1	7.8 ± 3.4	0.236 ^t^	11.9 ± 1.9	11.3 ± 2.1	0.306 ^t^
Monosomy of chromosome 3 (%)		10 (90.9)	8 (44.4)	0.019 ^P^	12 (86.2)	12 (46.2)	0.002 ^P^
Extrascleral extension (%)		4 (36.4)	10 (48.3)	0.450 ^F^	5 (17.2)	0 (0)	0.056 ^F^
Tumor cell type	spindle	1 (9.1)	8 (50.0)	0.117 ^F^	0 (0)	0 (0)	0.112 ^p^
	epithelioid	3 (27.3)	3 (18.8)		15 (57.7)	6 (33.3)	
	mixed	7 (63.6)	5 (31.3)		11 (42.3)	12 (66.7)	

* Quantitative data and qualitative data are expressed as the mean ± SD or median ± IQR and number of people (%), respectively; ^†^
*p* values refer to independent Student’s *t* test, Mann–Whitney U test, Pearson chi-square test and Fisher’s exact test used for exploring the differences in characteristics between two groups; t refers to independent Student’s *t* test; U refers to Mann–Whitney U test; P refers to Pearson chi-square test; F refers to Fisher’s exact test.

**Table 2 jcm-11-07224-t002:** Logistic regression analysis of hub genes affecting metastasis of the study patients.

		Univariate Analysis		Multivariate Analysis	
		OR (95% CI)	*p*	OR (95% CI)	*p*
GSE27831	ROBO1	1.002 (1.001–1.004)	0.010	1.002 (1.001–1.004)	0.010
	FMN1	0.997 (0.995–0.999)	0.013	0.998 (0.995–1.001)	1.197
	FYN	1.004 (1.000–1.008)	0.030	1.003 (0.995–1.011)	0.424
	FXR1	1.006 (1.001–1.010)	0.010	1.004 (0.999–1.009)	0.134
GSE22138	ROBO1	1.583 (1.173–2.136)	0.003	1.456 (1.054–2.011)	0.023
	FMN1	0.432 (0.260–0.719)	0.001	0.577 (0.336–0.992)	0.047
	FYN	2.491 (1.414–4.387)	0.002	1.804 (0.929–3.500)	0.081
	FXR1	2.009 (1.166–3.461)	0.012	1.078 (0.543–2.139)	0.830

Abbreviations: CI, confidence interval; OR, odds ratio.

**Table 3 jcm-11-07224-t003:** Individual area under the receiver-operating characteristic curve (AUC) and *p* values of hub genes to predict metastasis.

	GSE27831			GSE22138		
Genes	95% CI	AUC	*p*		AUC	*p*
ROBO1	0.788–1.000	0.904	<0.001	0.586–0.859	0.722	0.003
FMN1	0.678–0.978	0.828	0.003	0.662–0.897	0.780	<0.001
FYN	0.661–0.975	0.818	0.005	0.629–0.867	0.748	0.001
FXR1	0.697–0.990	0.843	0.002	0.570–0.830	0.700	0.007
ROBO1 combined FMN1				0.703–0.939	0.821	<0.001

Abbreviations: CI, confidence interval; AUC, area under the curve.

**Table 4 jcm-11-07224-t004:** The median metastasis-free survival times of the low-expression and high-expression groups of the hub genes.

		Low-Expression Group (Months)	High-Expression Group (Months)
GSE27831	ROBO1	31.0	41.5
	FMN1	44.0	23.0
	FYN	25.0	41.5
	FXR1	31.0	43.5
GSE22138	ROBO1	24.4	57.9
	FMN1	55.8	23.1
	FYN	24.5	58.7
	FXR1	24.5	55.8

**Table 5 jcm-11-07224-t005:** Prognostic factors affecting metastasis-free survival of the study patients.

		Univariate Analysis		Multivariate Analysis	
		HR (95% CI)	*p*	HR (95% CI)	*p*
GSE27831	ROBO1	0.999 (0.997–1.000)	0.009	0.999 (0.997–1.000)	0.009
	FMN1	1.002 (1.001–1.002)	0.002	1.000 (0.999–1.002)	0.455
	FYN	0.996 (0.993–0.999)	0.021	0.997 (0.993–1.001)	0.125
	FXR1	0.996 (0.993–0.999)	0.009	1.004 (0.995–1.002)	0.534
GSE22138	ROBO1	0.768 (0.644–0.915)	0.003	0.768 (0.634–0.931)	0.007
	FMN1	1.839 (1.346–2.512)	<0.001	1.832 (1.316–2.552)	<0.001
	FYN	0.605 (0.461–0.794)	<0.001	0.788 (0.570–1.090)	0.150
	FXR1	0.688 (0.511–0.928)	0.014	0.960 (0.645–1.430)	0.842

Abbreviations: CI, confidence interval; HR, hazard ratio.

## Data Availability

The datasets downloaded and analyzed during the current study are available on the GEO databases: GSE27831: https://www.ncbi.nlm.nih.gov/geo/query/acc.cgi?acc=GSE27831 (accessed on 2 May 2022); GSE22138: https://www.ncbi.nlm.nih.gov/gds/?term=GSE22138 (accessed on 2 May 2022). Informed consent was obtained from all subjects involved in the study.

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
