# Peer review of "Screening and Identification of Key Biomarkers in Metastatic Uveal Melanoma: Evidence from a Bioinformatic Analysis"

_jcm, 2022, doi:10.3390/jcm11237224_

Round 1
Reviewer 1 Report
The authors submitted a research article in which theymanaged to identify key biomarkers in the metastasis of uveal melanoma. They included 137 differentially expressed genes, consisting of 71 downregulated genes and 67 upregulated genes, and found that ROBO1 was an independent factor affecting metastasis and metastasis-free survival of UM, whereas ROBO1 and FMN1 were independent factors affecting metastasis and metastasis-free survival of UM in GSE22138. The aim of the study is clear and concise. The manuscript has a logical structure and composes off informative subsections that cover all aspects of the hypothesis. The tables a re legible. The conclusive part is informative and readable. Although the findings are impressive, I would like to put forward several issues to discuss.
1. Sample size calculation requires along with an example of esteemation.
2. Inter- and intra coefficients are needed to be discussed.
3. Validation of the morel. Please, give your opinion about it.
Author Response
Thank you for your review and comments of our manuscript. We are pleased to have the opportunity to address the concerns, and all amendments are indicated by red font in the revised manuscript.
We hope that this re-revised draft of the manuscript is acceptable for publication in Journal of clinical medicine and look forward to hearing from you soon.
- Sample size calculation requires along with an example of esteemation.
Response: Thanks for your review and comments. A server-based tool named SSizer1 was used to perform sample size calculation. The pilot data to determine the Sample Sufficiency was GSE1568772. All parameters were set as follows: the false discovery rate was 0.05, the cutoff of power value was 0.8, the cutoff of the area under the curve (AUC) was 0.8, the magnification of the simulated dataset compared with the pilot data was 1.2, the cutoff of accuracy was 0.7, and the cutoff of overlap was 0.5. 10 was determined to be the minimum sample size needed to satisfy the classification accuracy.
References
- Li F, Zhou Y, Zhang X, Tang J, Yang Q, Zhang Y, Luo Y, Hu J, Xue W, Qiu Y, He Q, Yang B, Zhu F. SSizer: Determining the Sample Sufficiency for Comparative Biological Study. J Mol Biol. 2020 May 15;432(11):3411-3421. doi: 10.1016/j.jmb.2020.01.027. Epub 2020 Feb 7. PMID: 32044343.
- Ness C, Katta K, Garred Ø, Kumar T et al. Integrated differential DNA methylation and gene expression of formalin-fixed paraffin-embedded uveal melanoma specimens identifies genes associated with early metastasis and poor prognosis. Exp Eye Res 2021 Feb;203:108426. PMID: 33387485.
- Inter- and intra coefficients are needed to be discussed.
Response: Thanks for your suggestion. As shown in Table 5, the multivariable Cox regression analysis in GSE27831 indicated that the regression coefficient of ROBO1 was -0.001, and the HR = 0.999. The higher expression of ROBO1, the higher risk of Metastasis will be. The multivariable Cox regression analysis in GSE22138 indicated that the regression coefficient of ROBO1 and FMN1 was -0.264 and 0.606, respectively. And the HR of ROBO1 and FMN1 was 0.768 and 1.832, respectively. The higher expression of ROBO1, the higher risk of metastasis will be. The higher expression of FMN1, the lower risk of metastasis will be. Similarly, as shown in Table 2, the results of the logistic regression analyses showed that ROBO1 was a protective factor against the occurrence of metastasis in GSE27831 and GSE22138. The higher expression of ROBO1 was able to reduce the probability of metastasis. And FMN1 was a risk factors factor for the occurrence of metastasis in GSE22138. The higher expression of FMN1 was able to improve the probability of metastasis. These conclusions can also be confirmed exactly in Figure 6.
- Validation of the morel. Please, give your opinion about it.
Response: Thanks for your suggestion. In our opinion, there can be two ways.
1) Two sample sets (training set and validation set) can be collected with matched baseline characteristics, including both metastatic and non-metastatic groups in each set, with at least 10 cases in each set. Gene expression and protein expression of ROBO1, FMN1, FYN, and FXR1 should be quantified, respectively. The prediction models should be built in the training set and then validated in the validation set.
2) UM tumor-bearing mice can be cultured. On the basis of the previous statistical analysis, key genes were selected. Gene-specific knockout and gene amplification of ROBO1, FMN1, FYN, and FXR1 were performed. Observing the changes in the degree of tumor metastasis and the metastasis-free survival in the test mice compared to control mice.
These are interesting works and deserve to be specifically studied further.
Reviewer 2 Report
Congratulation on your research. But, it would be very interesting if you analysed the expresion of hub genes in the samples that are high risc for metastatic disease ( eg. monosomy of chromosome 3...). Also compare obtained data with clinical and patohystological characteristics.
Author Response
Thank you for your review and comments on our manuscript. We are pleased to have the opportunity to address the concerns, and all amendments are indicated by red font in the revised manuscript.
We hope that this re-revised draft of the manuscript is acceptable for publication in Journal of clinical medicine and look forward to hearing from you soon.
- Congratulation on your research. But, it would be very interesting if you analysed the expresion of hub genes in the samples that are high risc for metastatic disease (eg. monosomy of chromosome 3...). Also compare obtained data with clinical and patohystological characteristics.
Response: Thanks for your suggestion. After completing this interesting work according to your instructions, we found some meaningful and interesting results. All results are in the attachment.
1) The expressions of ROBO1 and FXR1 in the samples with the characteristic of monosomy of chromosome 3 were lower than in other samples with significant differences in the GSE27831 (P=0.005 and P=0.007, respectively) and GSE22138 cohorts (both P<0.001). The expressions of FMN1 in the samples with the characteristic of monosomy of chromosome 3 were higher than in other samples with significant differences in the GSE27831 (P=0.018) and GSE22138 cohorts (P<0.001). The expressions of FYN in the samples with the characteristic of monosomy of chromosome 3 were lower than in other samples with a significant difference in the GSE22138 cohorts (P=0.001).
2) The expressions of ROBO1 and FXR1 in the samples with an age older than the mean age were higher than in other samples with significant differences in the GSE22138 cohorts (P=0.002 and P=0.042, respectively).
3) The expressions of FYN in the samples with tumor thickness greater than the mean tumor thickness were higher than in other samples with a significant difference in the GSE27831 cohorts (P=0.022).
4) The expressions of FYN and FXR1 in the samples with mixed tumor cell type were lower than in other samples with significant differences in the GSE22138 cohorts (P=0.046 and P=0.022, respectively).
These results are very interesting and deserve to be specifically studied further, for example, to further validate the impact of these clinical and pathological features on the expression of these genes and the mechanisms of influence.
